# MMP3 as a Molecular Link: Unraveling the Connection Between Ankylosing Spondylitis and Acute Coronary Syndrome

**DOI:** 10.3390/cells14080597

**Published:** 2025-04-15

**Authors:** Iliannis Y. Roa-Bruzón, Luis F. Duany-Almira, Yeminia M. Valle-Delgadillo, Héctor E. Flores-Salinas, Emmanuel Valdés-Alvarado, Jorge R. Padilla-Gutiérrez

**Affiliations:** 1Centro Universitario de Ciencias de la Salud, Instituto de Investigación en Ciencias Biomédicas (IICB), Universidad de Guadalajara, Guadalajara 44340, Jalisco, Mexico; iliannis.roa8556@alumnos.udg.mx (I.Y.R.-B.); luis.duany8455@alumnos.udg.mx (L.F.D.-A.); yeminia.valle@academicos.udg.mx (Y.M.V.-D.); emmanuel.valdes@academicos.udg.mx (E.V.-A.); 2Departamento de Biología Molecular y Genómica, Universidad de Guadalajara, Guadalajara 44340, Jalisco, Mexico; 3Especialidad en Cardiología, Unidad Médica de Alta Especialidad, Centro Médico Nacional de Occidente (CMNO), Departamento de Cardiología, Instituto Mexicano Del Seguro Social (IMSS), Guadalajara 44340, Jalisco, Mexico; 3nriqueflores@gmail.com

**Keywords:** ankylosing spondylitis, acute coronary syndrome, cardiovascular risk, chronic inflammation, extracellular matrix degradation, matrix metalloproteinase 3 (MMP-3)

## Abstract

Ankylosing spondylitis (AS) is a chronic inflammatory disease that primarily affects the joints, limiting patients’ mobility and quality of life. Recent studies have shown that patients with AS have a significantly higher risk of developing severe cardiovascular complications, such as acute coronary syndrome (ACS). A comprehensive review (2014–2024) included a study evaluating the significance of matrix metalloproteinase 3 (MMP-3) in cardiovascular risk among AS patients. The findings indicate that chronic inflammation in AS not only damages the joints but also contributes to the progression of cardiovascular diseases. At the molecular level, MMP-3 is instrumental in degrading the extracellular matrix, leading to instability in the atherosclerotic plaques and increasing the risk of ACS. Additionally, MMP-3 activation is related to the inflammatory pathways, such as tumor necrosis factor-alpha (TNF-α) and NF-κB, which amplify its effect on both joint destruction and vascular damage. This molecular approach offers new perspectives for understanding and treating AS and its cardiovascular complications, suggesting that MMP-3 inhibition could be a promising therapeutic strategy to mitigate cardiovascular risk in these patients.

## 1. Introduction

Ankylosing spondylitis (AS) and acute coronary syndrome (ACS) were once considered unrelated conditions. However, various studies have revealed a deeper connection between these two diseases, indicating that individuals with AS face a significantly increased risk of severe cardiovascular complications, such as ACS [1,2,3]. This link has become evident through epidemiological studies highlighting that the coexistence of AS with cardiovascular diseases, specifically coronary artery disease, is not an isolated phenomenon, but rather the result of shared pathophysiological mechanisms between both pathologies [4,5,6,7].

In addition to sharing mechanisms related to chronic inflammation, a higher prevalence of conventional cardiovascular risk factors has been observed in AS patients. Conditions such as metabolic syndrome and dyslipidemia are more common in this group, which amplifies their vulnerability to severe cardiovascular complications. This suggests that the persistent inflammation characteristic of AS, combined with these additional risk factors, significantly contributes to the intersection of both pathologies, exacerbating the impact of ACS in this patient group [5]. Although the relationship between AS and increased cardiovascular (CV) morbidity and mortality is well documented, controversy remains as to whether this increase in mortality is directly due to AS-specific cardiac manifestations or to a parallel rise in cardiovascular diseases associated with accelerated atherosclerosis that accompanies this chronic inflammatory disease [4].

To explore how matrix metalloproteinase 3 (MMP-3) influences this cardiovascular risk, a comprehensive review of articles published between 2014 and 2024 was conducted using databases such as PubMed, Elsevier, and other academic repositories. This review focused on the literature related to the impact of MMP-3 in AS and its connection with cardiovascular complications, aiming to examine how MMP-3 contributes to extracellular matrix degradation and interacts with inflammatory pathways that link AS to increased cardiovascular risk.

This underscores the need for the early identification and treatment of AS patients at cardiovascular risk, as well as to thoroughly explore the underlying molecular mechanisms connecting these two diseases. In this context, MMP-3 emerges as a central player, not only in the degradation of the extracellular matrix in AS [8,9,10] but also in the pathogenesis of atherosclerosis [11,12,13]. Therefore, a deeper understanding of the shared mechanisms between AS and ACS, including the involvement of MMP-3, could offer new opportunities to improve cardiovascular prevention and treatment in this patient population.

## 2. Clinical Evidence That Ankylosing Spondylitis Predisposes to Acute Coronary Syndrome

AS, or radiographic axial spondylarthritis, is a chronic, progressive, immunologic disease primarily affecting the axial skeleton, peripheral joints, entheses, and other areas such as the eyes and intestines. Unlike other forms of arthritis, AS is not associated with rheumatoid factors [14,15]. The etiology of AS remains not fully understood; however, its association with the human leukocyte antigen HLA-B27 has been extensively documented. Ankylosing spondylitis (AS) is the second most common cause of inflammatory arthritis globally [16]. The prevalence of AS closely correlates with the prevalence of HLA-B27 in different populations. Among individuals positive for this antigen, the prevalence of AS ranges between 5% and 6%. In the United States, HLA-B27 prevalence varies significantly across ethnic groups: 7.5% among non-Hispanic whites, 4.6% among Mexican Americans, and only 1.1% among non-Hispanic blacks, reflecting the differences in AS risk based on each group’s genetic makeup [17].

Globally, the estimated prevalence of AS among adult populations in North America and Europe ranges from 0.20% to 0.25%, while in populations with higher HLA-B27 prevalence, such as Arctic northern communities, it can reach 0.35%. In mainland China, military populations show a prevalence of 0.29%. In contrast, AS is extremely rare among indigenous populations of southern Africa, where HLA-B27 is absent [18]. Although its exact cause remains unknown, genetic studies are essential for a better understanding of the disease and its progression [19].

ACS, on the other hand, is a complex clinical entity encompassing a set of pathologies including unstable angina, non-ST-elevation myocardial infarction (NSTEMI), and ST-elevation myocardial infarction (STEMI), characterized by a sudden reduction in blood flow to the heart. These processes are primarily triggered by the instability, rupture, or erosion of atherosclerotic plaque, leading to coronary thrombosis with varying degrees of arterial occlusion. In cases like STEMI, the lack of blood flow results in the irreversible necrosis of heart tissue. Differentiating between ACS subtypes is key to effective diagnosis and treatment, given the variation in microcirculation impairment and myocardial damage [18,20].

Global mortality trends for ACS reflect significant disparities according to income levels and gender, although cardiovascular diseases remain the leading cause of global mortality, with an age-adjusted rate of 108.8 deaths per 100,000 people [21]. Men, especially those under 70, have shown higher mortality rates than women. In low-income regions such as Asia, Latin America, the Caribbean, and Africa, ACS mortality rates have shown minimal improvement over the past two decades. In contrast, high-income regions like North America, Europe, and Oceania have achieved significant reductions, exceeding 50%, thanks to better treatments and preventive measures. This shift highlights an epidemiological transition, where wealthier regions have seen substantial improvement, while advances in poorer regions have been minimal [22].

The relationship between AS and cardiovascular risk, particularly concerning ACS, has been the subject of several studies. Demonstrating a conclusive link between AS and cardiovascular diseases is challenging, as AS is a relatively rare condition, affecting approximately 1 in every 1000 people. Additionally, studies exploring this connection often involve small samples, limiting the precision and reliability of long-term results [4]. Despite this, compelling evidence suggests that AS not only affects the joints but also has a significant systemic impact, especially on the cardiovascular system. Chronic inflammation, a hallmark of AS, significantly contributes to the risk of severe cardiovascular events, including myocardial infarction and unstable angina [6,7].

Several studies have identified this connection between AS and ACS, indicating that systemic inflammation is a key factor. Bhattad et al. [4] analyzed a cohort of over 10,000 AS patients and concluded that these individuals had a significantly higher prevalence of coronary diseases. Compared to the general population, the ACS event rate in these patients was nearly double. This finding reinforces the idea that chronic inflammation in AS directly contributes to endothelial dysfunction and the progression of atherosclerosis. Similarly, Chou et al. [23] found a significantly higher incidence of severe coronary events, such as myocardial infarction, in AS patients compared to a control group. The coronary event rate among AS patients was 4.1 per 1000 person-years, compared to 2.6 in the general population. These results underscore the need to consider AS as an independent risk factor for ACS [24,25].

In addition to these studies, a meta-analysis conducted by Ungprasert et al. [5] corroborated the relationship between AS and coronary events. This analysis, which included multiple cohort studies, concluded that AS patients have a considerably higher risk of ACS, especially those who have lived with the disease for over 10 years. The prolonged duration of chronic inflammation appears to be a determining factor in the onset of severe cardiovascular complications.

Cardiovascular risk in AS patients also seems to be related to disease duration. Hung et al. [26] and Feng et al. [24] noted that patients aged 60 to 69 with long-standing AS have a particularly high risk of developing ACS. These findings underscore the importance of long-term monitoring in these patients, as persistent inflammation can significantly increase the risk of coronary events. As AS progresses, cardiovascular risk also increases, reinforcing the need for proactive disease management from the early stages [27].

An aspect that warrants special attention is the prognosis of AS patients after experiencing a coronary event. Södergren et al. [28] found that mortality in these patients after a first acute myocardial infarction is considerably higher than in the general population. This suggests that, in addition to being more prone to developing coronary events, AS patients also experience poorer outcomes after these events. Lai et al. [29] support this finding, noting that extra-articular manifestations, such as uveitis, can exacerbate the risk of infarction in AS patients. This emphasizes the importance of comprehensive disease management, not only in terms of joint health but also in monitoring systemic effects, including cardiovascular risks.

Another key point in the AS–ACS relationship is the impact of treatments. Karmacharya et al. [30] studied the effects of various anti-inflammatory treatments on cardiovascular risk in AS patients. While NSAIDs and TNF inhibitors were not associated with a significant increase in cardiovascular events, selective COX-2 inhibitors showed a trend toward reducing coronary event risk in AS patients. These findings suggest that the choice of anti-inflammatory treatment may be key in managing cardiovascular risk in this population [15,31].

The cardiovascular risk in young AS patients should not be overlooked. Although most studies focus on older patients, Wan et al. [32] documented a case of a young AS patient who suffered an acute myocardial infarction. This case highlights that cardiovascular risk in AS patients is not limited to the elderly. This suggests that cardiovascular monitoring should begin from the early stages of the disease, regardless of patient age, to prevent severe complications [17,33].

While several studies have indicated an increased risk of acute coronary syndrome (ACS) in ankylosing spondylitis (AS) patients, others have been unable to confirm these findings, suggesting possible limitations in the study design or the selection of analyzed populations. In the case of women with AS, an age-adjusted higher risk of developing ischemic heart disease has been observed. However, after adjusting results for non-steroidal anti-inflammatory drugs (NSAID) use, this association was reduced to a non-significant trend [25,27].

These results highlight the need for studies with a more robust approach that considers other important variables, such as disease duration or the presence of additional comorbidities. Furthermore, it would be interesting to explore how hormonal factors might influence the cardiovascular risk differences between men and women with AS, opening the door to more precise and tailored research in these subgroups [23,25].

While some studies have yielded mixed results, the overall evidence suggests a clear relationship between AS and the risk of severe cardiovascular events. Chronic systemic inflammation appears to be the primary underlying factor contributing to increased cardiovascular risk in this population. It is essential for physicians to consider these risks when treating AS patients, implementing regular cardiovascular monitoring and a comprehensive management strategy addressing both inflammation and cardiovascular risk factors [5,25,27].

## 3. Molecular Implications

### 3.1. MMP-3 in AS and ACS: Clinical and Pathological Aspects

Matrix metalloproteinase 3 (MMP-3) is fundamental to the inflammatory and tissue remodeling processes, establishing an important connection between AS and ACS. Both AS and ACS are chronic conditions affecting different systems; one targets the musculoskeletal system, while the other involves cardiovascular complications. However, both share inflammatory mechanisms and extracellular matrix remodeling where MMP-3 is a significant factor [9,11,12].

In ankylosing spondylitis (AS), chronic inflammation triggers a cascade of pro-inflammatory cytokines, such as tumor necrosis factor-alpha (TNF-α) [34] and interleukin-17 (IL-17) [35,36,37,38], which drive the expression of MMP-3. This metalloprotease, predominantly secreted by fibroblast-like synoviocytes derived from synovial fluid [39] in inflamed joints, plays a critical role in degrading the essential components of the extracellular matrix (ECM), leading to bone loss by disrupting the balance between bone formation and resorption [40]. Although the exact biological mechanisms underlying these processes remain incompletely understood [40], tissue degradation mediated by metalloproteinases such as MMP-3 is recognized as a key factor in the structural damage associated with AS. Several studies have demonstrated that elevated MMP-3 levels in AS patients are strongly correlated with increased disease activity and a faster progression toward ankylosis, or joint fusion, which significantly limits mobility and reduces quality of life [8,39].

In cardiovascular settings, MMP-3, predominantly secreted by cardiac fibroblasts and macrophages, is vital for the proteolytic breakdown of specific extracellular matrix components. Additionally, it facilitates the osteogenic transformation of vascular smooth muscle cells and contributes to medial artery calcification—processes that are significantly reduced in its absence [41]. In coronary arteries, elevated levels of MMP-3, induced by the chronic inflammation associated with AS, promote vascular inflammation and collagen degradation in atherosclerotic plaques. This degradation weakens the fibrous cap that stabilizes these plaques, thereby increasing the risk of plaque rupture, a critical event triggering acute coronary syndrome (ACS), which can result in myocardial infarction or unstable angina [12]. Therefore, the elevated activity of MMP-3 in AS not only contributes to joint damage but also facilitates the development of ACS through shared inflammatory mechanisms between these two conditions [9,12].

### 3.2. MMP-3 and Its Role in Extracellular Matrix Degradation

The human *MMP3* gene is located on the long arm of chromosome 11 (11q22.2), anti-sense strand, where other metalloproteinase genes like *MMP1, MMP7, MMP10, MMP8, MMP12, MMP13, MMP20*, and *MMP26* are also located. It consists of 7809 bases with 9 introns and 10 exons, coding for a 477-amino-acid protein, which is secreted from the cell as a 57 kD proenzyme [42,43].

Also known as stromelysin-1, MMP-3 is an enzyme integral to extracellular matrix remodeling and cell differentiation [40]. It significantly contributes to the breakdown of components such as fibronectin, laminin, various collagen types (III, IV, IX, X), and proteoglycans in cartilage [43]. Beyond its direct function in ECM degradation, MMP-3 can activate other metalloproteinases, such as MMP-1, MMP-7, and MMP-9, positioning it as a key regulator in the MMP activation cascade. Acting as an upstream activator, MMP-3 enhances tissue degradation processes, a mechanism especially significant in chronic inflammatory diseases, where uncontrolled tissue remodeling can result in excessive connective tissue breakdown, as observed in AS and atherosclerosis [9,12].

MMP-3 is synthesized as an inactive proenzyme, requiring activation through the plasmin cascade to perform its functions. Once activated, it acts on the ECM and participates in tissue remodeling under both physiological and pathological conditions. In some cases, its activation may occur intracellularly under specific stress conditions, as observed in dopaminergic neurons where the serine protease HTRA2 activates it—a process linked to neuronal degeneration in Parkinson’s disease [43].

Immunologically, MMP-3 contributes to innate immune responses and exhibits antiviral properties [43]. Evidence suggests that it may limit viral infections by enhancing the body’s antiviral immunity [44]. Moreover, inhibiting MMP-3 is emerging as a promising therapeutic approach for conditions such as COVID-19-associated acute respiratory distress syndrome (ARDS) [45], reflecting functions that extend beyond extracellular proteolysis to include the modulation of immune responses [44].

### 3.3. Molecular Pathways Linking AS and ACS

The TNF-alpha signaling pathway is essential for regulating inflammation, tissue damage, and cell death [46]. This pathway begins when TNF-α binds to its receptor TNFR1, triggering an intracellular signaling cascade involving proteins such as TRADD, TRAF2/5, and RIP1. These proteins subsequently activate the NF-κB and MAPK pathways, which are central to the regulation of inflammation and cell death [47,48].

In the NF-κB pathway, the transcription factor NF-κB remains inactive in the cytoplasm bound to IκBα. Upon TNFR1 activation, IκBα is phosphorylated and degraded, allowing NF-κB to translocate to the nucleus where it regulates the expression of pro-inflammatory and ECM-degrading genes, including MMP-3 [49]. In AS, chronic inflammation sustained by TNF-α and NF-κB results in MMP-3 overexpression in synovial cells and chondrocytes, thereby contributing to ECM degradation in joints [50,51,52,53,54]. This degradation leads to cartilage destruction, bone erosion, and eventually joint fusion—hallmarks of AS progression. Furthermore, chronic NF-κB activation may extend the effects of MMP-3 beyond the musculoskeletal system by contributing to endothelial dysfunction. The resultant systemic inflammatory state, along with elevated pro-inflammatory cytokines like TNF-α and IL-6, fosters an environment conducive to accelerated atherosclerosis in AS patients [50,55,56].

Beyond NF-κB, the mitogen-activated protein kinases (MAPK) pathway is instrumental in regulating MMP-3 expression. This pathway responds to inflammatory stimuli and cellular stress, with its activation contributing to MMP-3 production in both AS and atherosclerosis [55,57,58]. The MMP-3’s involvement in chronic inflammation is further connected to its interactions with pattern recognition receptors (PRRs), such as Toll-like receptors (TLRs). In AS, TLR activation in synovial and other immune cells promotes MMP-3 production along with other pro-inflammatory cytokines, perpetuating the cycle of inflammation and tissue destruction. Similarly, in atherosclerosis, TLR activation in endothelial cells and macrophages within atherosclerotic plaques induces MMP-3 expression, contributing to inflammation progression and plaque instability. Inhibiting PRRs or their downstream signaling pathways may thus represent a therapeutic strategy to reduce MMP-3 activation and mitigate the progression of both AS and atherosclerosis [52,53,58,59,60,61].

Additionally, tissue inhibitors of metalloproteinases (TIMPs) are essential for regulating MMP activity. Under normal conditions, TIMPs maintain a balance that prevents excessive ECM degradation. In AS, however, this balance is disrupted—resulting in MMP-3 overexpression and accelerated tissue degradation. A similar imbalance between MMPs and TIMPs is observed in ACS, where reduced TIMP levels and overexpression of MMP-3 contribute to plaque instability [62].

MMP-3-induced tissue remodeling in AS and ACS also shares common characteristics. In AS, MMP-3 degrades collagen and other ECM components in joints, leading to bone erosion and structural damage that limit patient mobility. In ACS, MMP-3 degrades collagen within the fibrous cap of atherosclerotic plaques, increasing the risk of plaque rupture and subsequent acute cardiovascular events [12,63,64].

Figure 1 shows the signaling pathways that regulate the expression and activation of MMP-3.

The TNF-α signaling pathway is also well documented in the pathogenesis of rheumatoid arthritis (RA), particularly in its contribution to synovial inflammation, cartilage destruction, and bone erosion [65]. The TNF-α/NF-κB-mediated pathway stimulates the overexpression of MMP-3 in synovial fibroblasts and chondrocytes, thereby facilitating extracellular matrix degradation [65,66]. This same mechanism, also described as active in ankylosing spondylitis (AS), is proposed to contribute to structural damage and endothelial dysfunction in both chronic inflammatory disorders.

The association between MMP-3 and the risk of cardiovascular events in RA patients has been examined from multiple perspectives. For instance, population-based and observational studies have shown that elevated levels of MMP-3 correlate with persistent inflammatory activity and structural damage [66,67], both of which are factors that increase cardiovascular disease risk. MMP-3 not only degrades the structural components of the joints but also participates in systemic processes, such as endothelial activation and the promotion of unstable atherosclerotic plaques.

In large-scale clinical studies, such as that by Chen et al. (2021) [68], RA patients were found to have an increased likelihood of developing major adverse cardiovascular events (MACE). Although the study focused on comorbidities and the impact of different antirheumatic drugs, the underlying role of chronic inflammation was evident as a key factor in cardiovascular risk. Similarly, the analysis by Meissner et al. (2023) [69], based on the RABBIT registry, reinforces this notion by demonstrating that effective inflammation control using biological DMARDs can reduce the risk of MACE, indirectly highlighting the relevance of the TNF-α–MMP axis in modulating cardiovascular damage.

In AS, although sustained TNF-α-induced MMP-3 activation and its contribution to joint damage are acknowledged, direct evidence of its role in MACE is less robust. While shared pathophysiological mechanisms such as systemic inflammatory pathway activation and MMP involvement in vascular dysfunction exist, they have not been documented with the same depth, nor are there comparative clinical studies that directly quantify the relative contribution of MMP-3 to MACE induction in AS versus RA.

Therefore, this review can and should be positioned as a pioneering contribution that proposes MMP-3 as a key molecular link between two clinical entities traditionally addressed separately: ankylosing spondylitis and acute coronary syndrome. This perspective not only enriches the field of biomedical knowledge but also raises direct implications for the development of new diagnostic and therapeutic strategies.

### 3.4. Therapeutic Potential of Targeting MMP-3

Since MMP-3 is closely associated with the progression of both AS and ACS, it has been proposed as a biomarker for evaluating cardiovascular event risk in AS patients. Studies have shown that AS patients with elevated blood levels of MMP-3 have a higher likelihood of developing subclinical atherosclerosis, thereby increasing the risk of severe cardiovascular events such as myocardial infarction. Similarly, in the context of ACS, elevated MMP-3 levels are associated with a worse prognosis and a higher risk of recurrent ischemic events [9,11,61].

The involvement of MMP-3 in ECM degradation in both conditions has spurred the proposal of its inhibition as a therapeutic strategy. Selective MMP-3 inhibitors could potentially slow cartilage degradation in AS, reducing disease progression and improving quality of life. In atherosclerosis, targeting MMP-3 may reduce plaque instability and decrease the risk of acute cardiovascular events. However, the development of effective MMP-3 inhibitors is challenging because MMPs are involved in essential physiological processes such as tissue remodeling and wound repair, and excessive inhibition could result in adverse effects [63,70,71].

Moreover, the interaction between MMP-3 and other MMPs is a significant area of research. MMP-3 not only directly degrades ECM components but also activates other MMPs, such as MMP-1 and MMP-9, thereby amplifying tissue degradation. In AS, this cascade contributes to accelerated joint tissue destruction, whereas in ACS, MMP-3’s activation of MMP-9 may exacerbate elastin degradation in coronary arteries, further increasing plaque instability and rupture risk [12,64].

Figure 2 illustrates the involvement of MMP-3 in the pathological interplay between AS and ACS.

## 4. Future Perspectives: MMP-3 Modulation

MMP-3 has attracted significant attention in the scientific community due to its fundamental involvement in extracellular matrix remodeling. Its modulation, either through specific inhibitors or combination therapies, offers new therapeutic perspectives to simultaneously address chronic inflammation and cardiovascular risk. The following are key strategies under exploration [9,11,12,40,61].

The development of specific MMP-3 inhibitors has been intensely explored in recent years, as these inhibitors block the enzyme’s catalytic activity, preventing uncontrolled extracellular matrix degradation that contributes to both joint destruction in AS and atherosclerotic plaque instability in ACS. Preclinical studies have shown that inhibiting MMP-3 can significantly reduce inflammation and tissue damage in animal models, supporting the potential of this approach in human diseases [62,68,70].

However, a key challenge remains selectivity. MMPs form a family of enzymes that perform critical functions in tissue homeostasis, and non-selective inhibition may interfere with normal repair processes. This issue has led researchers to focus on designing more specific inhibitors with an improved safety profile that can inhibit MMP-3 without negatively affecting other essential MMPs for tissue health [71,72].

Another promising strategy is combining MMP-3 inhibitors with biological therapies currently approved for ankylosing spondylitis. Tumor necrosis factor (TNF) inhibitors like infliximab and adalimumab have proven highly effective in reducing inflammation and AS symptoms. However, these treatments do not directly address MMP-3 activity, which remains a key factor in extracellular matrix degradation [71,72].

Combining MMP-3 inhibitors with these biological drugs could provide a more comprehensive therapy that controls systemic inflammation and prevents progressive tissue damage. This combination has the potential to reduce both active inflammation and tissue destruction, improving the long-term prognosis of AS patients [71,72].

Additionally, MMP-3 inhibitors could be combined with IL-17 inhibitors like secukinumab, which have shown a lower incidence of cardiovascular complications compared to TNF inhibitors. Since secukinumab blocks a different inflammatory pathway, its combination with an MMP-3 inhibitor may be particularly effective in reducing both inflammation and cardiovascular risk in AS patients at risk of ACS [73].

A comprehensive therapeutic approach could also involve combining MMP-3 inhibitors with treatments that address cardiovascular risk factors. Statins, known primarily for their ability to lower cholesterol levels, have shown anti-inflammatory effects and atherosclerotic plaque stabilization, making them an attractive option for patients with AS and at risk of acute coronary syndrome [74,75,76].

Combining statins with MMP-3 inhibitors could have a synergistic effect, stabilizing atherosclerotic plaques while reducing MMP-3’s destructive activity on the extracellular matrix. This would benefit patients in terms of joint inflammation control and lower their long-term cardiovascular risk [74,75].

Another promising combination is that of MMP-3 inhibitors with drugs like PCSK9 inhibitors, which have significantly reduced LDL levels in high-risk patients. By lowering cholesterol levels and stabilizing plaques, these drugs could complement MMP-3 inhibitors’ effects in cardiovascular protection [75].

Similarly, SGLT2 inhibitors, primarily used in diabetes treatment, have recently shown substantial cardiovascular benefits, including reduced heart failure and major cardiac events. Combining them with MMP-3 inhibitors could provide a comprehensive therapeutic approach addressing both inflammatory and cardiovascular risks in AS patients [74].

Another emerging research area is the genetic regulation of MMP-3. Recently, genetic polymorphisms that influence MMP-3 expression have been identified and may be associated with increased susceptibility to both ankylosing spondylitis and atherosclerosis. Modifying MMP-3 expressions at the genetic level, through gene therapy or RNA interference, could become a viable therapeutic strategy in the future [47,48,56].

Additionally, epigenetic regulation of MMP-3 is being explored as a potential intervention. Epigenetic modulators, which alter gene expression without directly modifying DNA, could be used to reduce MMP-3 expression in affected tissues. This approach has the potential to be more selective and long-lasting than chemical inhibitors, offering a long-term treatment option for patients with AS and high cardiovascular risk [53,77]. Table 1 summarizes the MMP-3 inhibitors and their specificity, providing a comprehensive overview of available agents

## 5. Challenges in MMP-3 Modulation

While MMP-3 inhibition holds considerable promise, it also presents several challenges. First, inhibitor specificity remains a key hurdle, as MMPs are essential for tissue repair. Non-selective inhibition could interfere with these processes and cause adverse effects, such as abnormal scarring or fibrosis. It is essential to develop inhibitors that exclusively target MMP-3 without affecting other enzymes in the family [67,69].

Moreover, clinical trials must be carefully designed to assess the benefits of reducing AS progression and its effects on cardiovascular health. Long-term studies are needed to measure not only improvement in joint symptoms but also cardiovascular outcomes, such as plaque stability and the incidence of cardiac events [9,36,40,67].

## 6. Conclusions

Chronic inflammation in AS not only deteriorates the joints but also significantly elevates the risk of ACS, with MMP-3 serving as an instrumental mediator in this connection. The activation of MMP-3 contributes to extracellular matrix degradation in both joint tissues and atherosclerotic plaques, promoting the instability of the latter and increasing the risk of severe cardiovascular events. This evidence positions MMP-3 as a potential biomarker and a promising therapeutic target, whose inhibition could offer an innovative approach to reduce systemic inflammation and mitigate vascular and joint damage. Combining MMP-3 inhibitors with current anti-inflammatory therapies, such as TNF-α or IL-17 inhibitors, could optimize inflammation control and reduce cardiovascular complications in AS patients. Additionally, new cell sequencing technologies and polymorphism analyses related to MMP-3 provide opportunities to understand its molecular regulation, paving the way toward more personalized treatments that address both AS progression and the associated cardiovascular risk. The findings suggest that MMP-3 inhibition, complemented by multidisciplinary approaches, could represent a significant advancement in the comprehensive management of AS and ACS, improving patients’ quality of life and reducing long-term cardiovascular morbidity.

## Figures and Tables

**Figure 1 cells-14-00597-f001:**
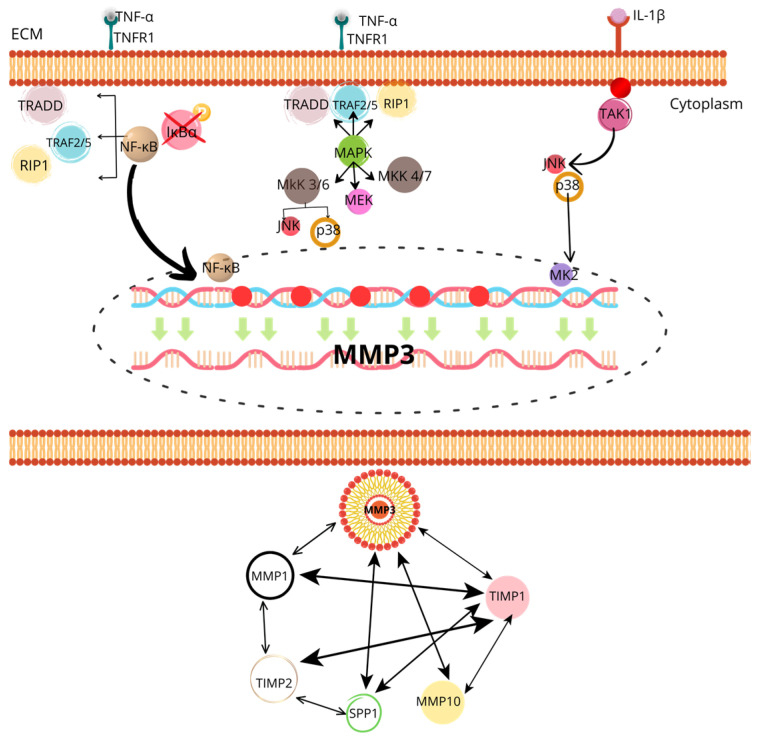
The image shows a detailed diagram of the signaling pathways that regulate the expression and activation of MMP-3 (matrix metalloproteinase 3), highlighting its role in extracellular matrix degradation and the activation of other MMPs. At the top, the activation of the NF-κB and MAPK pathways is observed through TNF-α and IL-1β signals, which bind to their respective receptors (TNFR1 for TNF-α and TAK1 for IL-1β) on the cell membrane. This activates signaling proteins such as TRADD, TRAF2/5, RIP1, and various kinases like MKK, MEK, and JNK/p38, which ultimately phosphorylate and activate NF-κB. This transcription factor translocates to the nucleus, where it promotes MMP-3 expression. At the bottom, MMP-3’s interaction with other metalloproteinases (MMP-1, MMP-10) and their tissue inhibitors (TIMP1, TIMP2) is shown, evidencing a complex regulatory system where MMP-3 not only acts directly but also modulates the activity of other MMPs. Additionally, interaction with SPP1 (extracellular matrix protein) involved in tissue degradation is observed, underscoring the central role of MMP-3 in inflammatory processes and tissue remodeling.

**Figure 2 cells-14-00597-f002:**
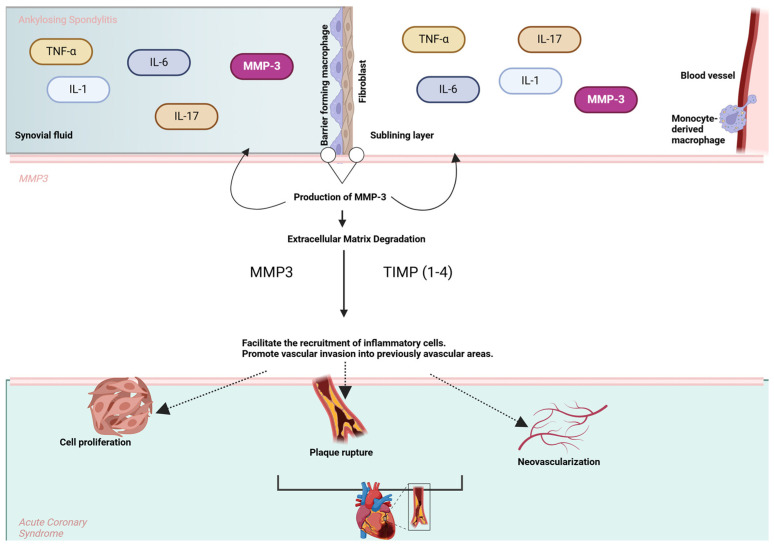
MMP-3 in the pathological link between AS and ACS. The upper section represents cytokines (e.g., TNF-α, IL-6, IL-1, and IL-17) involved in AS that stimulate the production of MMP-3 by macrophages, fibroblasts, and endothelial cells in the synovial lining. MMP-3 facilitates extracellular matrix (ECM) degradation and promotes inflammatory cell recruitment and vascular invasion. In the lower section, these processes contribute to key events in ACS, including plaque rupture, neovascularization, and cell proliferation, highlighting the shared inflammatory and tissue remodeling pathways between AS and cardiovascular complications.

**Table 1 cells-14-00597-t001:** MMP-3 inhibitors.

Group	Examples	Key Characteristics	Applications	Reference
Broad-Spectrum Inhibitors	Actinonin, PD166793, MMP Inhibitor V	Target multiple MMPs, including MMP-3; limited specificity	Potential for general MMP inhibition in inflammation-related conditions	(Kadry et al., 2021) [45]
Specific MMP-3 Inhibitors	MMP-3 Inhibitor VIII, MMP-3 Inhibitor V, UK 370106, UK 356618	Target MMP-3 with varying specificity; some are highly potent	More targeted therapeutic potential for MMP-3-driven pathologies	(Kadry et al., 2021) [45]
Peptide-Based Inhibitors	e(I), l(II)	Broad-spectrum inhibition, peptide-based, reduced cytotoxicity, enhanced bioavailability	Promising preclinical results for reversing inflammatory effects in osteoarthritis models	(Guarise et al., 2021) [78]

## Data Availability

No new data were generated in this study, and no human participants were involved.

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
