# Peer review of "MMP3 as a Molecular Link: Unraveling the Connection Between Ankylosing Spondylitis and Acute Coronary Syndrome"

_cells, 2025, doi:10.3390/cells14080597_

Round 1
Reviewer 1 Report
Comments and Suggestions for Authors
- Considering that chronic inflammation in AS contributes to the progression of ACS, this review article provides a detailed discussion on MMP-3 with epidemiological, clinical, and molecular data and suggests MMP-3 as a potential target molecule for the treatment of AS and ACS. At the same time, it is noteworthy that the article highlights the need to carefully consider the safety profile when considering the use of MMP-3 inhibitors.
- However, given the lack of clear evidence regarding the safety of MMP-3 inhibitors as monotherapy, proposing combination therapy with biologics or other agents requires great caution. The importance of safety and rigorous validation should be further emphasized.
- The resolution of the figures is too low and needs to be improved.
Author Response
Comments 1: Considering that chronic inflammation in AS contributes to the progression of ACS, this review article provides a detailed discussion on MMP-3 with epidemiological, clinical, and molecular data and suggests MMP-3 as a potential target molecule for the treatment of AS and ACS. At the same time, it is noteworthy that the article highlights the need to carefully consider the safety profile when considering the use of MMP-3 inhibitors.
Comments 2: However, given the lack of clear evidence regarding the safety of MMP-3 inhibitors as monotherapy, proposing combination therapy with biologics or other agents requires great caution. The importance of safety and rigorous validation should be further emphasized.
Response 2: We sincerely appreciate your valuable comments and the thoughtful review of our manuscript. Concerning the point on the safety of MMP-3 inhibitors as monotherapy and the cautious proposal of combining them with biologics or other agents, we thank you for this important remark. We believe the manuscript already emphasizes the need for rigorous safety validation, and the information presented is intended to serve as a scientific basis for future therapeutic strategies with improved efficacy and safety profiles.
Comments 3: The resolution of the figures is too low and needs to be improved.
Response 3: We sincerely appreciate your valuable comments and the thoughtful review of our manuscript. Lastly, in response to the comment about figure resolution, we will submit all images as separate high-resolution files to ensure optimal quality during the publication process.
Reviewer 2 Report
Comments and Suggestions for Authors
This is a pretty good narrative review regarding the role of bridging of MMP with development of cardiovascular events in ankylosing spondylitis. I have only one comment: the similar mechanism has been well documented in RA pathogenesis. The authors may add some data to compare the weighting of MMP in inducing MACE between RA and AS.
Author Response
Comments 1: This is a pretty good narrative review regarding the role of bridging of MMP with development of cardiovascular events in ankylosing spondylitis. I have only one comment: the similar mechanism has been well documented in RA pathogenesis. The authors may add some data to compare the weighting of MMP in inducing MACE between RA and AS.
Response 1: We sincerely appreciate your valuable comments and the thoughtful review of our manuscript. Regarding the suggestion to include a comparison of the role of MMPs in inducing cardiovascular events between rheumatoid arthritis (RA) and ankylosing spondylitis (AS), we have added further information between lines 274 and 304 of the manuscript.